# Identification of EGF Receptor and Thrombospondin-1 as Endogenous Targets of ER-Associated Degradation Enhancer EDEM1 in HeLa Cells

**DOI:** 10.3390/ijms241512171

**Published:** 2023-07-29

**Authors:** Kohta Miura, Riko Katsuki, Shusei Yoshida, Ren Ohta, Taku Tamura

**Affiliations:** 1Department of Life Science, Graduate School of Engineering Science, Akita University, Akita 010-8502, Japan; 2Department of Life Science, Faculty of Engineering Science, Akita University, Akita 010-8502, Japan

**Keywords:** endoplasmic reticulum-associated degradation (ERAD), EDEM1, ER chaperone, EGF receptor, TSP1

## Abstract

Secretory and membrane proteins are vital for cell activities, including intra- and intercellular communication. Therefore, protein quality control in the endoplasmic reticulum (ER) is an essential and crucial process for eukaryotic cells. Endoplasmic reticulum-associated degradation (ERAD) targets misfolded proteins during the protein maturation process in the ER and leads to their disposal. This process maintains the ER productive function and prevents misfolded protein stress (i.e., ER stress). The ERAD-stimulating factor ER degradation-enhancing α mannosidase-like 1 protein (EDEM1) acts on misfolded proteins to accelerate ERAD, thereby maintaining the productivity of the ER. However, the detail mechanism underlying the function of EDEM1 in ERAD is not completely understood due to a lack of established physiological substrate proteins. In this study, we attempted to identify substrate proteins for EDEM1 using siRNA. The matrix component thrombospondin-1 (TSP1) and epidermal growth factor receptor (EGFR) were identified as candidate targets of EDEM1. Their protein maturation status and cellular localization were markedly affected by knockdown of EDEM1. We also showed that EDEM1 physically associates with EGFR and enhances EGFR degradation via ERAD. Our data highlight the physiological role of EDEM1 in maintaining specific target proteins and provide a potential approach to the regulation of expression of clinically important proteins.

## 1. Introduction

In eukaryotes, secretory and membrane proteins are important for a variety of intra- and intercellular events. Endoplasmic reticulum (ER) is the organelle involved in the production of secretory and membrane proteins. In the ER, organized protein quality control systems, productive protein folding, and selective degradation of misfolded proteins are executed. Protein folding and assembly in the ER are assisted by ER-resident chaperones, disulfide bond-producing enzymes, and N-linked glycan modification enzymes. Moreover, terminally misfolded or unassembled proteins are disposed through a process termed ER-associated degradation or ERAD. These processes, which complement each other, are termed ER quality control systems (ERQC) [1]. Co- and post-translational N-glycosylation, as well as its trimming, are vital signs for protein maturation and ERAD [2,3]. As protein biosynthesis proceeds, mannose residues of N-glycan on glycoproteins are removed and act as an ERAD indicator in the ER [4]. The ER is a membrane-surrounded tubule structure filled with various proteins. Thus, maturation of newly translated proteins is a complicated process, leading to failed protein products. Exceeding the capacity of ERAD causes accumulation of misfolded proteins at the ER and triggers stress responses. This process results in the production of folding support proteins (e.g., molecular chaperones or modifying enzymes, ERAD-related factors), and finally leads to apoptosis [5].

Recent studies on ERQC have highlighted the physiological importance of ERAD. Regulation of protein turnover by ERAD is required for cellular development and physiological balances [6,7]. The SEL1L/Hrd1 complex is a central mechanism of ERAD where misfolded proteins are gathered, retro-translocated to the cytoplasm, and polyubiquitylated for proteasomal degradation [8]. It has been shown that the SEL1L/Hrd1 complex is required for various life events and cellular functions, including the development and growth regulation of the liver [9,10,11]. OS-9, which recognizes 5–6 mannose N-glycan of ERAD substrates and acts as a downstream factor of SEL1L/Hrd1 ERAD pathway, targets CD147 and the Na-K-2Cl co-transporter NKCC2 [3,12,13]. Recent proteomic investigations uncovered endogenous substrates of specific ERAD factors and allowed the elucidation of the detailed mechanism underlying their interaction with target proteins [14,15]. Thus, the identification of endogenous target proteins for ERAD in tissues or in developmental stages extends the role and function of protein degradation in physiology and development.

EDEM1, an ER-associated degradation-enhancing alpha mannosidase-like factor 1, is an ERAD-enhancing factor that facilitates the clearance of various ERAD model proteins [16]. It has been demonstrated that EDEM1 accelerates several steps of ERAD. EDEM1 exhibits mannose-trimming activity of N-glycans and presents misfolded proteins following interaction with downstream ERAD factors, such as OS-9 and XTP3B [17,18,19]. During ERAD acceleration, EDEM1 selectively binds to immature proteins that contain free-thiol or a hydrophobic region exposed at the molecular surface [19,20]. Furthermore, the mannosidase-like domain of EDEM1 is required for association with SEL1L in a mannose-trimming-dependent manner, suggesting that the role of EDEM1 in ERAD is to present substrate misfolded proteins to the SEL1L/Hrd1 complex [20]. In addition, EDEM1 interacts with transmembrane-form and soluble ERAD substrates, as well as ERAD-related cellular factors [21]. This is achieved by creating both soluble and transmembrane forms of EDEM1 through slow signal sequence cleavage [21]. However, the physiological significance of EDEM1 and EDEM1-mediated ERAD remains unclear owing to the lack of information on the targets of EDEM1.

In this study, we attempted to identify endogenous substrates of EDEM1 and explored their role in ERAD. Downregulation of EDEM1 gene expression by siRNA induced the accumulation of certain ER-resident chaperones. This finding suggested that inhibition of EDEM1-mediated ERAD causes accumulation of misfolded proteins in the ER. We ascertained that thrombospondin-1 (TSP1) and epidermal growth factor receptor (EGFR) are endogenous targets of EDEM1 for ERAD in HeLa cells. Furthermore, we characterized their degradation under basal state and EDEM1-knockdown (KD) conditions. We found that their cellular location and folding status were affected by EDEM1-mediated ERAD. We also revealed that EDEM1 associates with EGFR and enhances the degradation of EGFR that is inhibited by the proteasome inhibitor, MG132. These data suggested that ERAD is involved in EGFR turnover. Collectively, our results suggest that basal expression of EDEM1 is required for maintaining the productivity of the ER through the downregulation of misfolded ER proteins, including TSP1 and EGFR.

## 2. Results

### 2.1. Downregulation of EDEM1 Gene Expression Induced ER Stress and Apoptosis

It is well established that the protein expression levels of EDEM1 greatly affect the disposal of misfolded ER proteins through ERAD. EDEM1 gene expression is upregulated by ER stress in which unfolded protein responses (UPR) restrict the production of secretory and membrane proteins and activate ERAD. However, the activity and role of EDEM1 in cells with low ER stress are not well understood. To explore the function of EDEM1 at basal levels, we performed transient mRNA downregulation of EDEM1 using siRNA. The EDEM1 mRNA levels were clearly decreased compared with those of the housekeeping gene glyceraldehyde-3-phosphate dehydrogenase (GAPDH) (Figure 1A). Moreover, it was found that the gene expression levels of the spliced form of XBP1 (XBP1s), an ER stress marker and transcription factor involved in UPR, were markedly enhanced by EDEM1 KD (Figure 1A). We examined whether EDEM1 KD by siRNA is applicable to exogenously expressed ERAD substrates null Hong Kong alpha1 antitrypsin (NHK) and CD10 C143Y [22]. The degradation of both ERAD substrates was suppressed or even aggregated by siRNA for EDEM1 (Appendix A), indicating that our experimental condition is effective in HeLa cells. When the treatment with siRNA for EDEM1 was extended up to 96 h, the protein expression of CCAAT-enhancer-binding protein homologous protein (CHOP) (an UPR marker that stimulates ER stress-mediated apoptosis [23]) was upregulated (Figure 1B). Moreover, typical morphological features of apoptosis, such as cell shrinkage and nuclear fragmentation, were also observed (Figure 1C). Consequently, these results suggest that the functions of EDEM1 (even at basal levels) are to maintain the ERQC and prevent ER stress by ERAD of misfolded proteins.

### 2.2. ER Chaperone Clustering by EDEM1 siRNA KD

Indirect immunostaining using antibodies against ER-resident proteins was conducted to visualize the effect of EDEM1 KD on ER morphology. A soluble ER heat shock protein 70 (HSP70) family chaperone-binding immunoglobulin protein (BiP) exhibited as an ER network structure under siRNA to luciferase as a negative control (Figure 2a). However, after EDEM1 KD, BiP appeared as punctuated structures (Figure 2d). Such morphological change was also observed for ERp19, one of the protein disulfide isomerase-family (PDI-family) proteins that assists in disulfide bond formation of ER-folding proteins [24] (Figure 2b,e). Indirect immunostaining using an anti-KDEL antibody, which recognizes proteins bearing the C-term KDEL sequence, such as BiP, calreticulin (CRT), PDI, and ERp57, exhibited punctuated structures similar to those noted with anti-BiP (Appendix A). This result suggests that misfolded proteins destined for ERAD were assembled following EDEM1 KD and caused the accumulation of the above ER chaperones. Interestingly, calnexin (CNX), a membrane-anchored homolog of CRT, and UDP-glucose:glycoprotein glucosyltransferase 1 (UGGT1) did not form a cluster after EDEM1 KD (Appendix A). Our morphological analyses suggested that EDEM1 targets misfolded proteins after the CNX cycle and proteins with insufficient disulfide bonds. UGGT1 recognizes immature glycoproteins and acts as a gatekeeper of ER glycoproteins to add monoglucose to the N-glycan of immature proteins for re-entry into the CNX cycle [25]. The differences in the obtained immunostaining observations may result from the distinction between two ER protein-folding mechanisms, namely CNX-UGGT1 and CRT-ERp57-BiP; the former targets newly synthesized proteins, while the latter targets relatively misfolded proteins.

### 2.3. EDEM1 KD Resulted in ER Accumulation of TSP1

Next, we sought to understand the importance of EDEM1 in cell viability. Therefore, we explored secretory or membrane proteins that are degraded by EDEM1-mediated ERAD. Such proteins may accumulate in the ER by EDEM1 KD probably due to insufficient disposal from the ER and aggregate owing to the hydrophobicity of the molecules. Through biochemical analysis, the secretory protein TSP1 and the single transmembrane protein EGFR were identified as the endogenous substrate proteins of EDEM1 for ERAD. TSP1 is a soluble extracellular matrix protein that mediates intercellular connection through heparin binding and an EGF-like module [26]. Post-translational modifications of TSP1, including several disulfide bonds and N-glycans, are important for its structural stability and function [27]. Indirect immunostaining showed that, in terms of intracellular location, TSP1 exhibited an ER-like pattern and showed similar distribution to ERp57 (Figure 3A(a–c)). Following EDEM1 gene suppression, ERp57 exhibited similar clustering to that observed for BiP and ERp19 (see Figure 2a,d). Notably, some punctuations of TSP1 were colocalized with ERp57 (Figure 3A(d–f)). These results suggested that the EDEM1-mediated ERAD could remove misfolded TSP1 from the ER. To further characterize the protein aggregation status, we biochemically analyzed intracellular TSP1 under EDEM1 gene KD condition. Misfolded proteins tend to expose the hydrophobic region to the molecular surface of the proteins. Thus, the detergent-insoluble fraction of the cell lysate was analyzed to evaluate the folding status. KD of EDEM1 increased the insoluble fraction of TSP1 (Figure 3B,C). These results are consistent with those obtained from the indirect immunostaining assay that showed punctuation of TSP1. These findings suggested that ERAD by EDEM1 is important for the clearance of misfolded TSP1 molecules from the ER.

### 2.4. EDEM1 KD Induced EGFR Accumulation in ER

EGFR directs cell proliferation through EGF-mediated extracellular signaling. In addition, it is regarded as an oncogene and its gene expression is upregulated in various types of cancer [28]. Since EGFR contains numerous disulfide bonds and N-linked glycans, we tested whether EGFR is a target of EDEM1 for ERAD. EGFR was almost undetectable in the detergent-insoluble fraction under the negative control siRNA. These data suggested that endogenous EGFR is properly folded and misfolded EGFR is degraded smoothly in HeLa cells (Figure 4A, lane 4). However, detergent-insoluble EGFR was detected after EDEM1 KD (Figure 4A, lanes 5 and 6). The amount of aggregated EGFR was correlated with the efficacy of EDEM1 KD (comparing #1 and #2. See Figure 1), indicating that EDEM1 works for the disposal of misfolded EGFR during its folding in the ER (Figure 4B).

To ascertain whether EDEM1 is involved in EGFR biosynthesis, we constructed hemagglutinin-tagged (HA-tagged) EGFR at the C-term (EGFR-HA). Thereafter, the intracellular trafficking of EGFR-HA during siRNA experiments was examined. Indirect immunostaining with antibodies against HA and KDEL (ER marker) showed that EGFR-HA was localized at the plasma membrane under the negative control siRNA (Figure 4C(a)). This result indicated that exogenously expressed EGFR-HA was properly folded in the ER and transported to the plasma membrane in HeLa cells. However, EGFR-HA was merged with anti-KDEL staining and exhibited some clusters under EDEM1 KD (Figure 4C(d–f)). Furthermore, EGFR-HA was also localized to the nuclear membrane, which is continuous with the ER (Figure 4C(d)). These results suggested that EDEM1 is involved in misfolded EGFR clearance that is retained in the ER by the ERQC. To further confirm the involvement of EDEM1 in the intracellular trafficking of EGFR-HA, we performed endoglycosidase H (Endo H) digestion of the N-glycans of EGFR-HA. Endo H treatment resulted in both slower (Endo H-resistant form) and faster (Endo H-sensitive form) migration of EGFR-HA, representing post-Golgi and ER localization, respectively. Following treatment with the negative control siRNA, markedly higher levels of the Endo H-resistant form of EGFR versus the Endo H-sensitive form were detected (Figure 4D, lane 2). However, KD of EDEM1 increased the levels of Endo H-sensitive EGFR-HA (comparing lanes 2 and 5, Figure 4D). These results suggested that ERAD by EDEM1 clears ER-accumulated EGFR-HA and is important for the proper cellular localization of EGFR-HA.

### 2.5. EDEM1 Facilitated EGFR Degradation through ERAD

To ascertain whether EDEM1 participates in EGFR degradation through ERAD, EGFR-HA was co-transfected with C-terminal Flag-tagged EDEM1 and the expression levels of EGFR-HA were examined. As the levels of EDEM1-Flag were increased, the protein expression of EGFR-HA was decreased (Figure 5A,B). We also sought to investigate whether this decrease is attributed to the enhancement of ERAD of EGFR-HA by EDEM1. Hence, proteasome activity was inhibited using MG132 during the transfection, and EGFR-HA degradation was verified. EDEM1 co-expression reduced the protein expression of EGFR-HA by approximately 50% compared with the control (Figure 5C, comparing lanes 1 and 2). However, in the presence of MG132, EGFR-HA expression was recovered to approximately 75% (Figure 5C,D). These results suggested that the effect of EDEM1-Flag overexpression on EGFR-HA downregulation is due to the upregulation of ERAD. Cell surface expression of EGFR-HA was observed following the overexpression of EDEM1-Flag (Appendix A). Thus, EDEM1 could participate in the biosynthesis of misfolded EGFR-HA, but not in that of matured EGFR-HA. We investigated the physical interaction of EDEM1-Flag with EGFR-HA by co-immunoprecipitation experiments (Figure 5E). Immunoisolation and western blotting analyses revealed that EDEM1-Flag was associated with EGFR-HA (Figure 5E, lane 8). Taken together, our results suggested that EDEM1 contributes to the turnover of ERQC substrate proteins, including TSP1 and EGFR, by ERAD to prevent ER stress in cells.

## 3. Discussion

The removal of terminally misfolded proteins from the ER is crucial for cell viability. This is because the accumulation of such misfolded proteins causes ER stress. Prolonged ER stress leads to apoptosis. Therefore, cells invoke UPR to recover from ER stress by enhancing protein folding and ERAD. EDEM1 is a gene induced by UPR that reinforces the degradation of misfolded proteins through ERAD. In this study, we showed that EDEM1 KD by siRNA induces the upregulation of XBP1s (an established UPR marker), and longer siRNA treatment leads to apoptosis. ER stress is associated with the development of numerous diseases, including breast cancer [28]. Hence, the management of ER stress in vivo offers promise for the treatment or prevention of such diseases.

The present analyses identified two clinically important molecules, namely TSP1 and EGFR, as the endogenous targets of EDEM1 for ERAD in HeLa cells. As both TSP1 and EGFR are intricated molecules, protein maturation of these molecules in the ER would be relatively difficult. It has been revealed that TSP1 forms a disulfide bond-mediated trimer and consists of structurally complex domains [29]. TSP1 is a matrix protein that can manage cell–cell communication and is required for angiogenesis. Therefore, it functions as a prognostic marker, particularly in breast cancer [30]. Research studies have highlighted the functional role of TSP1 in the development of diseases, including cancer [31,32].

EGFR is a well-established predictive biomarker in cancer. It is widely accepted that the overproduction of EGFR and mutation of the EGFR gene strongly affect several types of cancer [28]. Protein turnover and lysosomal degradation of EGFR after activation by binding of the extracellular ligand EGF has been well documented [33]. However, the protein maturation process of EGFR in the ER and the subsequent membrane trafficking to the plasma membrane have not been adequately addressed. Moreover, at present, there is limited information regarding the formation of the disulfide bonds of EGFR, the type of isomerases in the ER, and the type of ER-resident chaperone required for EGFR folding and assembly. A comprehensive analysis using a mass spectrometry-based capturing assay revealed that EDEM1 is an EGFR-associated protein [34]. This analysis supports our results, suggesting the physiological interaction of EGFR with EDEM1 and EDEM1-mediated degradation of EGFR by ERAD. EDEM1 acts on misfolded proteins only during the maturation process in the ER. Thus, EDEM1 may exert a limited effect on the regulation of EGFR or TSP1 protein expression. Nevertheless, our results may assist in understanding the mechanisms underlying the effects of these clinically important molecules on protein biosynthesis and maturation.

Silencing of the EDEM1 gene suppresses the degradation of different ERAD substrates [35,36,37,38]. Of note, we found that the aggregation and turnover of lysosomal-associated membrane protein 1 (Lamp1) and Lamp2a (lysosomal transmembrane proteins) were not significantly affected by EDEM1 KD (Appendix A), though it led to the aggregation of EGFR and TSP1 (Figure 3, Figure 4 and Figure 5). These findings suggested that EDEM1 is endowed with substrate specificity for ERAD. The three isoforms EDEM1, EDEM2, and EDEM3 differ in tissue distribution, expression levels, and membrane topology; EDEM1 is a soluble and type II membrane-anchored form, while EDEM2 and EDEM3 are soluble [21,39,40,41]. Thus, EDEM1-3 should be assigned ERAD substrate specificity since the ERAD pathway involves various secretory and membrane proteins.

It has been demonstrated that some secretory or membrane proteins require specific ER proteins for maturation or ERAD. Using such proteins as a model substrate, the precise role and mechanism of ERQC has been explored. ER-resident glucosidases UGGT1 and UGGT2 recognize and reglucosylate mal-folded secretory/membrane proteins to be returned into the lectin chaperone folding process termed the CNX cycle [25]. Endogenous targets that contain characteristic structural motifs for UGGT1 and UGGT2 have been previously identified [42]. The HSP90-like ER chaperone GRP94 targets γ-aminobutyric acid type A (GABAA) receptors as an endogenous ERAD substrate [43]. In addition, OS-9 (an ERAD component associated with the SEL1L/Hrd1 complex [44]) is the target of GRP94-mediated ERAD [45]. In the present study, we propose a novel aspect of EDEM1; at basal levels (i.e., not induced by UPR), EDEM1 acts on EGFR and TSP1 for ERAD. Analysis using endogenously but not exogenously expressed proteins provides a more natural aspect of the productive protein pathway and ERAD in cells and in vivo. The combination of an inductive approach and comprehensive analyses could be effective for understanding the detailed ERQC mechanism and role in physiology.

## 4. Materials and Methods

### 4.1. Antibodies, Materials, and Plasmids

Rabbit polyclonal anti-BiP, anti-ERp57, anti-CNX, and anti-CRT antibodies were kindly provided by Dr. Tetsuro Yamashita (Iwate University, Morioka, Japan). Rabbit polyclonal anti-CHOP (15204-1-AP; Proteintech Japan, Tokyo, Japan) and rabbit monoclonal anti-EGFR (#4267; Cell Signaling Technology Japan, Tokyo, Japan) antibodies were purchased. Mouse monoclonal anti-ERp19 (sc-376410; Santa Cruz Biotechnology, Dallas, TX, USA), anti-UGGT1 (sc-374565 (H-9); Santa Cruz Biotechnology, Dallas, TX, USA), anti-TSP1 (sc-393504 (C-8); Santa Cruz Biotechnology, Dallas, TX, USA), anti-Lamp1 (H4A3; Santa Cruz Biotechnology, Dallas, TX, USA), anti-Lamp2a (H4B4; Santa Cruz Biotechnology, Dallas, TX, USA), anti-αTubulin (T6074; Sigma-Aldrich, St. Louis, MO, USA), anti-Flag (F1804; Sigma-Aldrich, St. Louis, MO, USA) anti-HA (M180-3; MBL Japan, Tokyo, Japan), anti-KDEL (M181-3; MBL Japan, Tokyo, Japan), and rat anti-HA (7C9; Proteintech Japan, Tokyo, Japan) antibodies were also purchased. For transfection and cellular protein expression analysis, plasmid vectors were generated as follows. The pCX4-bsr-EDEM1-Flag expressing C-terminal Flag-tagged EDEM1 was subcloned from the pEDEM1-3XFlag vector [20] through standard polymerase chain reaction (PCR) and cloning methods into the HpaI and NotI sites of pCX4-bsr. pCMV-EGFR-HA expressing C-terminal HA-tagged EGFR was generated by standard reverse transcription-PCR (RT-PCR) and PCR. Plasmids encoding NHK-Turquoise were provided by Dr. Ikuo Wada (Fukushima Medical University, Fukushima, Japan). Plasmids expressing CD10 C143Y-Flag were previously cloned and characterized [22]. The primers used for DNA cloning are included in Appendix A.

### 4.2. Cell Culture, Transfection, siRNA, and RT-PCR

HeLa cells used in our previous study [22] were cultured in Dulbecco’s modified Eagle’s medium containing 10% fetal bovine serum and 1% penicillin/streptomycin in an incubator at 37 °C, with 5% CO_2_ and constant humidity. The cells were transfected with Fugene HD (E2312; Roche Diagnostics K.K., Tokyo, Japan) according to the instructions provided by the manufacturer. The siRNA experiment was performed with RNAiMAX (677592; Invitrogen, Waltham, MA, USA) for 72 or 96 h according to the instructions provided by the manufacturer. Gene expression was verified by RT-PCR using reverse transcriptase Superscript II (Nippongene, Tokyo, Japan) and DNA polymerase Phusion (New England Biolabs, Ipswich, MA, USA) based on standard protocols. The primers used for RT-PCR and siRNA in this study are shown in Appendix A.

### 4.3. Cell Lysis, Pulldown Experiments, and Western Blotting

Proteins were extracted with 1% Triton X100-containing lysis buffer: 50 mM HEPES (pH: 7.5); 150 mM NaCl; 10 mM glucose; 2 units/mL hexokinase (Sigma); and protease inhibitor cocktails. Detergent-soluble and insoluble fractions were separated by centrifugation at 18,000× *g* at 4 °C for 10 min. Protein concentration of the soluble fraction was determined by a standard Bradford assay system. For pulldown experiments, the cell lysate was treated using anti-Flag agarose beads (Sigma), as previously described [22]. Protein samples were subjected to reducing SDS-polyacrylamide gel electrophoresis (SDS-PAGE) and subsequently transferred onto a polyvinylidene difluoride membrane (Millipore, Burlington, MA, USA). The membrane was blocked with tris-buffered saline containing 5% skim milk and 0.05% Tween 20 (Nacalai, Kyoto, Japan) at 4 °C overnight. Next, the proteins were detected using indicated primary antibodies and secondary antibodies. Finally, they were visualized with enhanced chemiluminescence and the ChemiDoc system (Bio-Rad, Hercules, CA, USA). Densitometric analysis was performed using the ImageJ software version 1.53 (National Institute of Health, Bethesda, MD, USA). The expression of proteins of interest was normalized to that of αTubulin. The error bars in figures represent the standard deviation for at least three independent experiments.

### 4.4. Glycosidase Digestion

Treatment of denatured cellular proteins with Endo H and peptide-N-glycosidase F (PNGase F), both purchased from New England Biolabs, was conducted as previously described [46]. Total cellular protein extracted using the aforementioned lysis buffer was denatured in the presence of Nonidet P-40. Subsequently, the samples were resolved by SDS-PAGE and analyzed using western blotting.

### 4.5. Indirect Immunostaining

HeLa cells were seeded in a 35-mm dish. This was followed by transfection or siRNA treatment as indicated in the figure legends. After fixation using 4% paraformaldehyde in phosphate-buffered saline (PBS; Wako, Osaka, Japan) at room temperature for 10 min, cells were incubated with 0.1% Triton X-100 in immunostaining buffer (PBS containing 1% goat serum (Gibco ThermoFisher Scientific, Waltham, MA, USA) and 5% glycerol) at 2 °C for 1 min for membrane permeabilization. Fixation using methanol was conducted as follows. After the addition of ice-cold methanol and incubation at −20 °C for 10 min, the transfected cells were washed with PBS and blocked with the immunostaining buffer at room temperature for 5 min. Next, the cells were incubated with indicated primary antibodies and secondary antibodies, and stained with 4′,6-diamidino-2-phenylindole (DAPI; Wako) at room temperature for 5 min. The coverslip was washed with distilled water, placed on the slide, and fixed with Mowiol (Sigma). Confocal imaging was conducted with the LSM780 microscope (Zeiss, Jena, Germany).

## Figures and Tables

**Figure 1 ijms-24-12171-f001:**
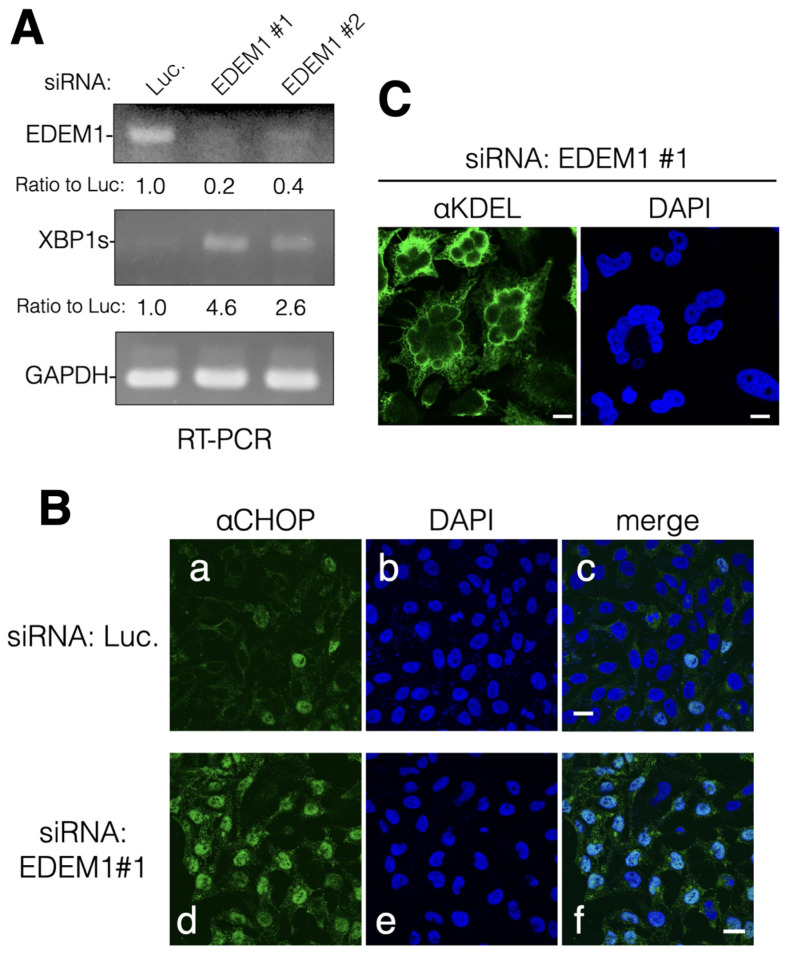
siRNA-mediated mRNA downregulation of EDEM1 caused ER stress and apoptosis. Knockdown of EDEM1 gene by siRNA upregulated the UPR and induced apoptosis in HeLa cells. (**A**) After treatment with indicated siRNA (luciferase [Luc.]), EDEM1#1 and EDEM1#2 for 72 h), HeLa cells were lysed, and mRNA samples were prepared as described in the Materials and Methods section. The mRNA of EDEM1, XBP1s, and GAPDH of the control cells (Luc.) and EDEM1 KD cells was amplified and detected by agarose gel electrophoresis. Each band was quantified using NIH ImageJ software and normalized by GAPDH. The ratio was calculated with Luc being 1.0 and noted at the bottom. (**B**) After 72 h of siRNA treatment, HeLa cells were fixed with paraformaldehyde and immunodetected using an anti-CHOP antibody (an ER stress marker associated with apoptosis, panel (**a**,**d**). DAPI (**b**,**e**) and merged images (**c**,**f**) are shown. Scale bars in panel c and f represent 10 µm. (**C**) After 96 h of siRNA treatment, HeLa cells were fixed and immunodetected using an anti-KDEL antibody that recognizes ER-resident KDEL sequence-containing proteins. Scale bar represents 10 µm.

**Figure 2 ijms-24-12171-f002:**
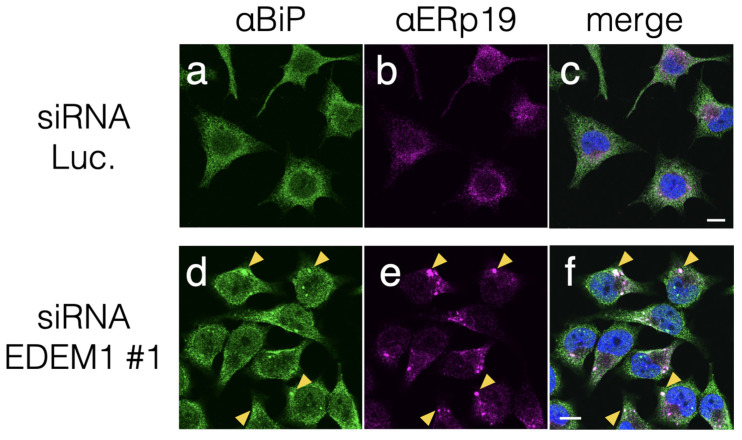
EDEM1 KD induced ER chaperone accumulation. Treatment with EDEM1 siRNA induced the accumulation of some ER-resident chaperones in HeLa cells. After the control (Luc., (**a**–**c**)) or EDEM1#1 (**d**–**f**) siRNA treatment, ER chaperone proteins BiP (**a**,**d**) and ERp19 (**b**,**e**) were immunodetected using corresponding antibodies. Merged images after DAPI staining are also shown (**c**,**f**). Some co-localization points in panels (**d**–**f**) are indicated by yellow triangles. Scale bars in panel (**c**,**f**) represent 10 µm.

**Figure 3 ijms-24-12171-f003:**
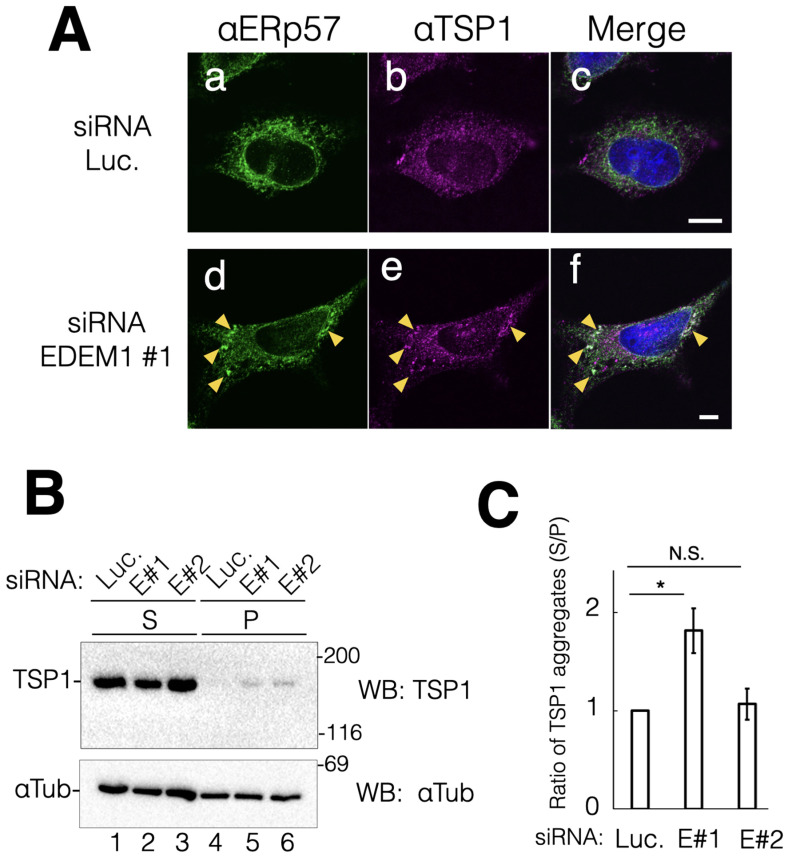
Downregulation of EDEM1 induced intracellular aggregation of TSP1. EDEM1 KD led to the accumulation of the matrix soluble protein TSP1 in cells. (**A**) After the control or EDEM1#1 siRNA treatment, ER-resident chaperone ERp57 (**a**,**d**) and TSP1 (**b**,**e**) were immunodetected using corresponding antibodies. Merged images after DAPI staining are also shown (**c**,**f**). Co-localization points of ERp57 and TSP1 in EDEM1 KD cells are indicated by yellow triangles (**d**–**f**). Scale bars represent 10 µm. (**B**) siRNA-treated cells were lysed and separated into soluble supernatant (S) and insoluble precipitation (P) fractions. Both TSP1 and αTubulin were detected by western blotting with indicated antibodies for the control (lanes 1 and 4) and EDEM1 KD cells (lanes 2 and 5, 3 and 6). Lane numbers are shown at the bottom of the gel. The molecular weight of marker proteins (kDa) is shown at the right side of the gel. (**C**) Data in (**B**) were quantified. αTubulin was used as the cellular loading control. In each sample set (lanes 1 and 4, 2 and 5, 3 and 6, respectively), the values of anti-Flag bands were normalized to those of anti-αTubulin. Notably, the values of soluble bands were normalized to those of precipitation bands. Asterisks refer to the control sample, indicating statistical significance (* *p* < 0.05; N.S., not significant) according to Student’s *t*-test. Data represent the mean ± standard deviation of three independent experiments.

**Figure 4 ijms-24-12171-f004:**
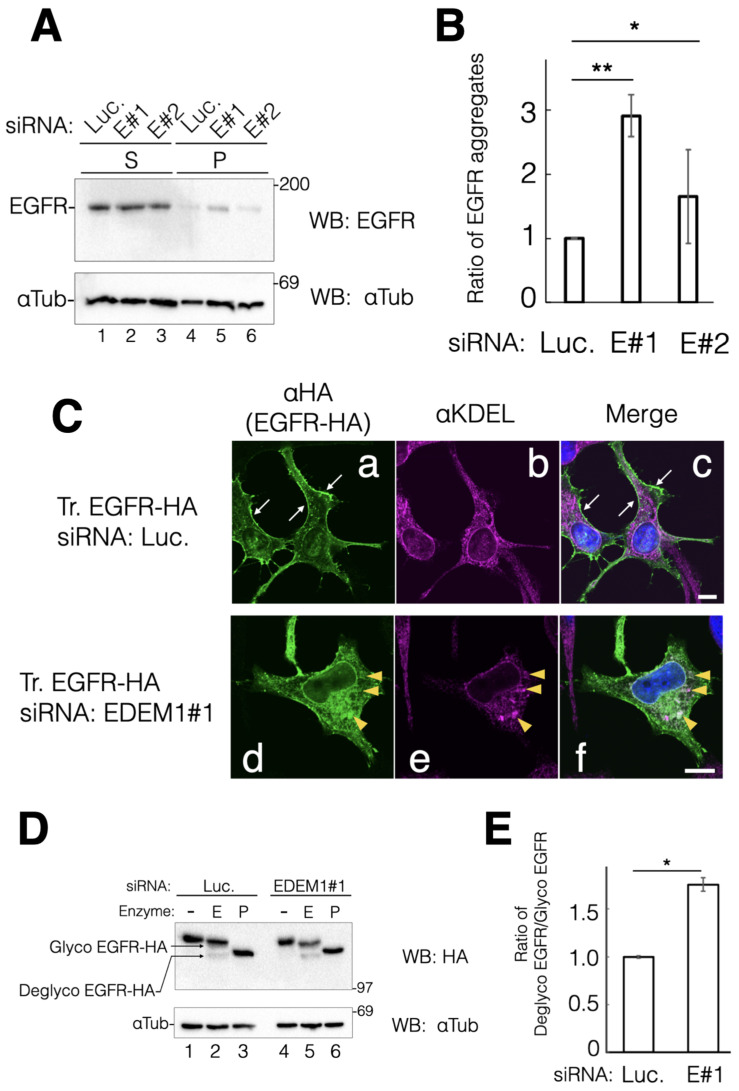
EDEM1 knockdown (KD) increased the detergent-insoluble and intracellular EGFR fractions. EDEM1 KD affected the cellular localization and protein solubility of EGFR. (**A**) siRNA-treated cells were lysed and fractionated into detergent-soluble (S) and -insoluble fractions (P). Protein bands of EGFR and αTubulin were immunodecorated with indicated antibodies for the control (lanes 1 and 4) and EDEM1 KD (lanes 2 and 5, 3 and 6). Lane numbers are shown at the bottom of the gel. The molecular weight of marker proteins (kDa) is shown at the right side of the gel. (**B**) Quantification of (**A**). αTubulin was used as the cellular loading control. In each sample set, (lanes 1 and 4, 2 and 5, 3 and 6, respectively), the values of anti-Flag bands were normalized to those of anti-αTubulin. Of note, the values of soluble bands were normalized to those of precipitation bands. Asterisks refer to the control sample, indicating statistical significance (* *p* < 0.05; ** *p* < 0.01) by Student’s *t*-test. Data represent the mean ± standard deviation of three independent experiments. (**C**) During siRNA treatment with the control (**a**–**c**) or EDEM1#1 (**d**–**f**), cells were transfected with EGFR-HA, and cellular localization was verified by indirect immunostaining. After fixation with methanol, proteins were visualized using anti-HA (EGFR-HA, green) and anti-KDEL (KDEL motif-containing proteins, magenta) antibodies. Merged images after DAPI staining are shown as (**c**) (control) and (**f**) (EDEM1#1). Plasma membrane localization of EGFR-HA in (**a**) and (**c**) is indicated by white arrows. The co-localization of EGFR-HA and KDEL proteins is represented by yellow triangles. Scale bars in (**c**,**f**) represent 10 µm. (**D**) Ratio of post-Golgi and ER-retained EGFR-HA following EDEM1 KD. Denatured lysates from HeLa cells treated with the control (lanes 1–3) or EDEM1#1 (lanes 4–6) siRNA were digested with mock (lanes 1 and 4), Endo H (lanes 2 and 5), and PNGase F (lanes 3 and 6). The positions of N-glycosylated (Glyco EGFR-HA, post-Golgi) and deglycosylated EGFR-HA (Deglyco, ER) in SDS-PAGE are indicated at the left side of the gel. (**E**) Quantification of data from (**D**) is shown. In lanes 2 and 5, the values of deglyco EGFR-HA bands were divided by those of glyco EGFR-HA bands. The values were standardized to those of αTubulin. Asterisks refer to the control sample, indicating statistical significance (* *p* < 0.05) by Student’s *t*-test. Data represent the mean ± standard deviation of three independent experiments.

**Figure 5 ijms-24-12171-f005:**
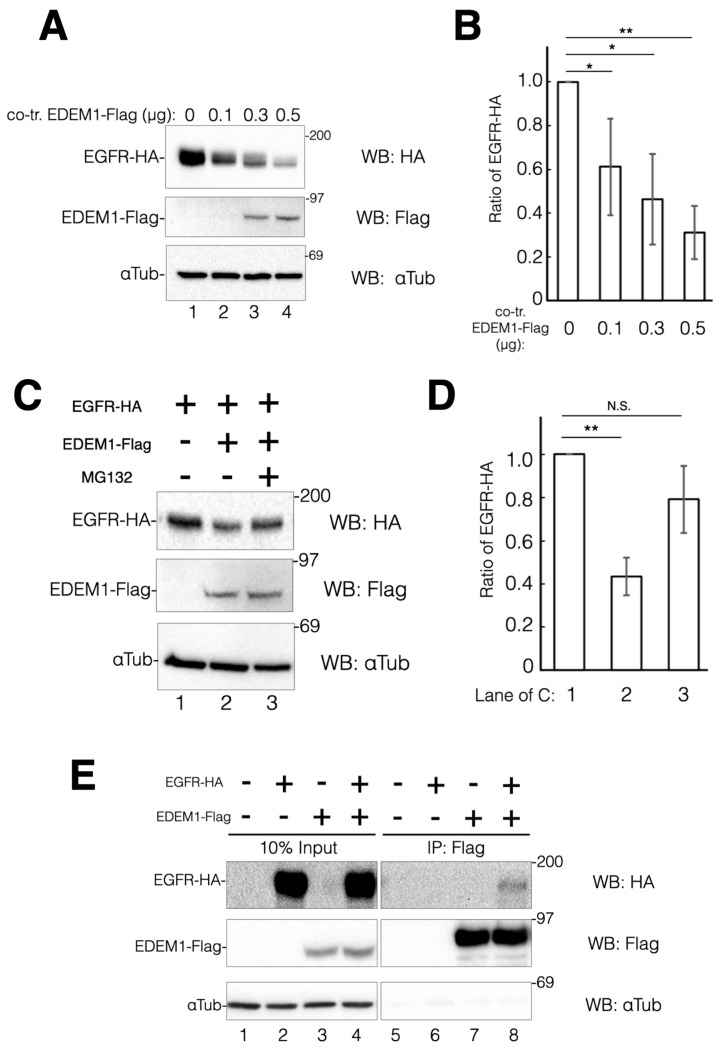
Co-expression of EDEM1 promoted EGFR degradation by ERAD. EGFR protein turnover is accelerated by overexpression of EDEM1 through the ERAD pathway. (**A**) Turnover of EGFR-HA is dependent on the co-expression of EDEM1-Flag. HeLa cells were transfected with a fixed amount of EGFR-HA and various EDEM1-Flag coding plasmids as indicated. Proteins were detected by western blotting using indicated antibodies. (**B**) Quantification of (**A**). The amount of EGFR-HA was calculated using αTubulin as a cellular control. Asterisks refer to the control sample, indicating statistical significance (* *p* < 0.05; ** *p* < 0.01) by Student’s *t*-test. Data represent the mean ± standard deviation of three independent experiments. (**C**) Downregulation of EGFR-HA by EDEM1-Flag is mediated by the proteasome. HeLa cells were transfected with EGFR-HA (lane 1) or EGFR-HA and EDEM1-Flag (lanes 2 and 3). In co-transfected cells, cells were treated with MG132 (lane 3). Proteins were detected by western blotting with indicated antibodies. (**D**) Quantification of (**C**). Protein bands in (**C**) were quantified and plotted. Asterisks refer to the control sample, indicating statistical significance (N.S., not significant; ** *p* < 0.01) by Student’s *t*-test. Data represent the mean ± standard deviation of three independent experiments. (**E**) Physiological interaction of EDEM1-Flag and EGFR-HA. HeLa cells were co-transfected with EDEM1-Flag and EGFR-HA and part of the cell lysate (10%) was loaded as input (lanes 1–4), while the remaining lysate was used for immunoprecipitation for anti-Flag (lanes 5–8). Proteins were detected using indicated antibodies. In co-transfected cells, interaction of EDEM1-Flag and EGFR-HA was observed (lane 8).

## Data Availability

Data are available from the corresponding author upon request.

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
