# Peer review of "Identification of EGF Receptor and Thrombospondin-1 as Endogenous Targets of ER-Associated Degradation Enhancer EDEM1 in HeLa Cells"

_ijms, 2023, doi:10.3390/ijms241512171_

Round 1

Reviewer 1 Report

With the work entitled “Identification of EGF receptor and thrombospondin-1 as endogenous targets of ER-associated degradation enhancers EDEM1 in HeLa cells” the authors described the role of EDEM1 in the function of the ERAD mechanism to control the degradation of defective proteins and/or with negative regulation. These data are important for understanding the basic mechanisms and identifying new targets for EDEM1 in the proposed model, opening new perspectives.

However, many problems with the figures were identified, despite being in the text and well referenced.

1. First, the attached figures could be better organized and with a better resolution. The indication of some figures is missing in the document attached to the manuscript. For example, in the document ijms-2468127-original-images the second page looks like figure 3, but it is a continuation of figure 1.

2. Why was a qRT-PCR not performed in Figure 1? Semiquantitative RT-PCR should be normalized by endogenous. There is not even a densitometry showing the siRNA inhibition fold of EDEM1, let alone XBP1s. Authors should perform a real time PCR or demonstrate the variation between points by densitometry.

3. Figures 1B, 1C, including the whole of figure 2 are not attached to the original figures file for evaluation. How to evaluate?

4. Where is figure 3A?

5. In the images of WBs referring to anti-alpha-Tubulin in figure 4B there is no marking in lanes 4-6. Why, since it is an endogenous protein?

6. Where is figure 4C?

7. The original images of the 4D figure are changed, they do not refer to the edited images.

8. What is the percent transient transfection efficiency of the siRNAs used?

In short, I cannot finish all the analyzes due to the absence of some images and figures.

Reviewer 2 Report

Authors Kohta Miura et al studied the impact of EDEM1 extinction on the ER-associated degradation (ERAD).

To this purpose, they used only one models: 1- siRNA silencing EDEM1 in the HeLa cell line in vitro.

Under knock down conditions, they observed up-regulation of XBP1 splicing, of CHOP protein expression and morphological features of apoptosis.

By IF, they observed that some ER chaperone proteins (BIP, calreticulin (CRT), PDI, TSP1 and ERp57) exhibited punctuated structures. By WB analysis of the detergent-insoluble fraction of KD Hela cells, they observed the enrichment of TSP1 in the insoluble fraction.

By co-expression of EGFR-HA and EDEM1-Flag, they observed a correlation between the up-regulation of EDEM1 and the down-regulation of EGFR, that was partially recovered by inhibiting the proteasome activity (addition of MG132).

This work fits in the scope of the section “Biochemistry” of the journal.

The question is about the role of EDEM1 in ERAD.

As the experiments are done only in 1 cell line, without any confirmation in primary cells, and than the data are only correlatives observations, I think that the relevance is too poor to be published in your journal.

Major comments:

1.     The data observed in this work are only correlatives.

2.     The data are produced in Hela Cell line.

3.     The Figures are missing.

Minor comments:

English language could be improved.

English language could be improved.

Round 2

Reviewer 1 Report

The authors modified some parts of the text and the presentation of figures. The files uploaded by the system are messed up. The image file ijms-2468127-non-published contains the figures that were disorganized and missing. However, in the ijms-2468127-original-images file, the figures are still confused, missing some parts and shuffled.

I would like to see the final figures organized into their panels, where I could correlate with the text of the manuscript. Only then can I come to a final conclusion.

Ok

Author Response

To the reviewer #1,

Thanks for your patient and close review for our manuscript.

According to recomendation of the reviewer #1, we added figure images to the manuscript. Supplemental materials are shown after acknowledgment section all together. And we also added some instructions to the text describing Fig. 2 (section 2.2, highlighted in red). We hope that this new version of our manuscript is suitable for publication.

Round 3

Reviewer 1 Report

After the suggested revisions and modifications, the authors were able to organize the manuscript in a presentable way.

Author Response

We are pleased to receive a positive comment from the reviewer #1 as "After the suggested revisions and modifications, the authors were able to organize the manuscript in a presentable way". Please let me know if there is anything else we should fix in the manuscript.